# The Regulatory Effects of Citrus Peel Powder on Liver Metabolites and Gut Flora in Mice with Non-Alcoholic Fatty Liver Disease (NAFLD)

**DOI:** 10.3390/foods10123022

**Published:** 2021-12-06

**Authors:** Meiyi Hu, Li Zhang, Zheng Ruan, Peiheng Han, Yujuan Yu

**Affiliations:** 1Beijing Advanced Innovation Center for Food Nutrition and Human Health, Beijing Technology and Business University (BTBU), Beijing 100048, China; humeiyi2345@163.com; 2State Key Laboratory of Food Science and Technology, Institute of Nutrition and School of Food Science, Nanchang University, Nanchang 330047, China; zhangli@ncu.edu.cn (L.Z.); Hanpeiheng06@163.com (P.H.); juan2035@163.com (Y.Y.)

**Keywords:** citrus peel, liver metabolites, gut flora, non-alcoholic fatty liver disease, pomelo, orange

## Abstract

*Gannan* navel orange and *Jinggang* pomelo, belonging to the genus Citrus, are good sources of phenolic compounds, which are mainly concentrated in the peel. These phenolic compounds are considered promising in the prevention and treatment of non-alcoholic fatty liver disease (NAFLD). In order to maximize nutrients retention and bioactivity in the peel, pomelo peel and orange peel were processed using freeze-drying technology and mixed in the ratio (pomelo peel powder 50% and orange peel powder 50%) to make citrus peel powder (CPP). The purpose of this study was to explore new strategies and mechanisms associated with the consumption of CPP to alleviate nonalcoholic fatty liver injury, lipid metabolism disorders, and gut microbiota dysbiosis in obese mice induced by high-fat diet (HFD). The results showed that after 12 weeks of CPP administration, CPP supplementation had a strong inhibitory effect on HFD-induced weight gain, hepatic fat accumulation, dyslipidemia, and the release of pro-inflammatory cytokines. In particular, CPP modulates the composition of the intestinal flora, such as increasing the relative abundance of phylum *Firmicutes*, genus *Faecalibaculum*, genus *Lactobacillus*, genus *Dubosiella*, and genus *Lachnospiraceae_NK4A136_ group* and decreasing the relative abundance of phylum *Bacteroidota*, genus *Helicobacter*, and genus *Bacteroides*. These results suggest that CPP has a preventive effect on NAFLD, which can be related to the regulation of intestinal flora.

## 1. Introduction

Currently, non-alcoholic fatty liver disease (NAFLD) has become the most common liver disease worldwide [1]. NAFLD is a metabolic disease that manifests as a buildup of lipids in liver cells but without alcohol abuse or other indeterminate liver damage factors [2]. Notably, NAFLD is a serious threat to human health, and with delayed treatment, NAFLD will gradually deteriorate and evolve into non-alcoholic steatohepatitis (NASH), cirrhosis, and even liver cancer [3]. NAFLD occurs for a variety of reasons, such as high-fat diet, sedentary lifestyle, genetically induced hepatic lipid accumulation, and insulin resistance [4]. Clinical and scientific studies have shown lifestyle changes as the most basic treatment for NAFLD, including proper meal planning and exercise [5,6]. The tendency of patients to not comply with the designed plan makes behavioral interventions difficult to achieve in most cases. In addition, medications for the treatment of NAFLD remain controversial. Therefore, potent natural products should be considered for the treatment of NAFLD.

Citrus fruits, belonging to the *Rutaceae* family, are one of the most abundant fruits in the world [7]. However, citrus fruits that are processed each year produce large amounts of residues. It is worth noting that citrus peel, as the main citrus fruits residue, will lead to serious environmental pollution if not further utilized [8]. Citrus peel is reported to be a good source of phenolic compounds, pectin, and essential oil [9]. *Jinggang* pomelo (*Citrus grandis* (L.) Osbeck) and *Gannan* navel orange (*Citrus sinensis* Osbeck cv. Newhall), both belonging to the citrus genus, have attracted much attention due to the rich phenolic compounds of their fruit peel. Tocmo et al. [10] and Razola-Diaz et al. [11] found that the phenolic compounds in pomelo peel and orange peel were mainly composed of mostly flavonoids and a small amount phenolic acids and that the biological activities exhibited by both peels, such as antioxidant, hypoglycemic, hypolipidemic, anti-inflammatory, and anticancer activities, were mainly attributed to their phenolic compounds content. Moreover, many researchers have demonstrated that a diet rich in polyphenols is effective in the prevention and treatment of NAFLD [12,13]. Bayram et al. [14] suggested that polyphenols can improve NAFLD through different mechanisms, including antioxidant effects, anti-inflammatory effects, improved insulin sensitivity, prevented β-fatty acid oxidation, reduced de-novo lipogenesis, and improved adipokine. Hence, it was implied that pomelo peel and orange peel had a protective effect on NAFLD.

Liver disease is associated with altered intestinal flora, and intestinal microecological balance plays a major role in the prevention and treatment of NAFLD [15,16]. Mounting evidence suggests that intestinal flora is important in lipid metabolism and nutrient and energy acquisition and may be a cause of metabolic diseases [17,18,19]. To date, targeting intestinal flora through dietary interventions may be one of the most promising strategies to mitigate metabolic disease [20,21]. Natural products have been reported to alter the composition of intestinal flora, thereby regulating lipid metabolism and preventing obesity [22,23,24]. However, there are no reports on the mechanism of action of pomelo peels and orange peels together for the treatment of NAFLD and the effect on intestinal flora. Therefore, in this study, pomelo peel and orange peel were powdered using low-temperature freeze-drying technique and mixed in the ratio (pomelo peel powder:orange peel powder = 1:1) to form a citrus peel powder (CPP). The purpose of this research was to investigate the effects of CPP on liver fat accumulation, obesity, liver inflammation, and intestinal flora composition in high-fat-diet (HFD) rats. This study can provide a scientific basis for the prevention of NAFLD through dietary supplementation of citrus fruit by-products.

## 2. Materials and Methods

### 2.1. Materials and Chemicals

*Jinggang* pomelo (*Citrus grandis* (L.) Osbeck) and *Gannan* navel orange (*Citrus sinensis* Osbeck cv. Newhall) were purchased from a Farmers Market of Ji’an city, Jiangxi Province, China. The two kinds of peels are made into a powder by vacuum freeze-drying technology and were mixed according to the ratio (pomelo peel powder: orange peel powder = 1:1) to make citrus peel powder (CPP).

All phenolic standards (purity > 98%) for UPLC-MS/MS analysis were purchased from Aladdin Biochemical Tech Co., Ltd., (Shanghai, China). Acetonitrile and formic acid of mass spectrometry (MS) grade and all chemicals of analytical grade were obtained from Sinopharm Chemical Reagent Co., Ltd., (Beijing, China).

### 2.2. Determination of the Bioactive Components of CPP

The phenolic components of CPP were analyzed by ultra-high performance liquid chromatography-triple quadrupole mass spectrometry (UPLC-MS/MS) coupled with electrospray ionization source (ESI) (Waters Corporation, Milford, MA, USA) mass spectrometry using the method of Tang et al. [25] with some modifications. Chromatographic separation was performed on a chromatographic column (21 × 50 mm, Waters ACQUITY UPLC BEH C18). The mobile phase consisted of (A) 0.25% formic acid aqueous solution and (B) 0.25% formic acid acetonitrile solution. The gradient was carried out as 0–20 min, 95–80% B; 15–25 min, 80–70% B; 25–40 min, 70–0% B; 40–45 min, 0–95% B; and 45–55 min, 95% B for equilibrium. The mobile phase flow rate was kept at 0.3 mL/min, while temperature was 35 °C. The sample (1 μL) was injected to the system and PDA detector. MS (Waters Corporation, Milford, MA, USA) were obtained at positive and negative ion modes. The source parameters were as follows: ESI source voltage of 2.0 kV, desolventizing temperature 500 °C. The cone hole voltage was 30 V for positive ion mode and 30 V for the negative ion mode. Full scans MS was measured from *m*/*z* 100 to 1000. The metabolites were identified based on chromatographic elution times, chemical composition, MS/MS cleavage patterns, and comparisons with existing standards and references.

Since studies have shown that citrus fruit peels also contain other bioactive components, such as essential oils, pectin, and limonene [9]. Therefore, we extracted and determined the essential oil in CPP using the hydrodistillation method and determined the content of limonene in it using gas chromatography-mass spectrometry (GC-MS) (Shimadzu, Japan) [26]. The pectin in CPP was extracted according to the method of Wandee et al. [27], and the pectin yield was determined.

### 2.3. Animals, Diet, and Experimental Design

All animal experimental procedures used in this study (Figure 1) were reviewed and approved by the Institutional Animal Ethical Committee Nanchang University (Permission number: SYXK2019-0021). The animal experimental design was based on the method provided by Yuan et al. [28] with slight modifications. Twenty-four 6-week-old Sprague–Dawley rats were bought from Tian Qin Biotechnology (Changsha, China). All animals were fed in a controlled environment, with temperature 24 ± 1 °C, humidity 40–60%, and 12-h daylight cycle. Then, the rats were randomly divided into 3 groups (*n* = 8): normal control group (ND), HFD-diet group (HF), and a HFD united by gavaging the homogenate of CPP (HFC). The rats in ND group were fed with low-fat chow (15%, 20%, and 65% of total calories from fat, protein, and carbohydrate, respectively). The rats in the HF group were fed HFD (60%, 20%, and 20% of total calories from fat, protein, and carbohydrate, respectively). In addition, the rats in ND and HF groups received an equal volume of phosphate-buffered saline (PBS). The rats in the HFC group were fed a HFD accompanied by daily gavage with 1 g/kg·BW CPP every day. Food and water were unlimited for all rats during 12-week experiment period.

The body weights of all rats were weighed and recorded weekly throughout the experiment. After 12 weeks of feeding, all rats were fasted overnight. All rats were euthanized by carbon dioxide anesthesia followed by cervical dislocation. All blood specimens were cardiac punctured, and serum was collected by centrifugation. Subsequently, all serum and fecal samples were loaded into separate EP tubes and stored in a −80 °C refrigerator. The liver, epididymis fat, perirenal fat, and subcutaneous fat were collected intact for weighing and recording and stored at −80 °C.

### 2.4. Histological Analysis

The method used was described as Guo et al. [29] with some modifications. The fresh liver and adipose tissues were fixed in 4% paraformaldehyde for 24 h, then dehydrated and fixed in a paraffin, and 6-µm-thick sections of tissues were cut and stained with hematoxylin and eosin (H&E) for the histological analysis. The pathological morphology was observed a light microscope (XD-202 inverted biological microscope, Nanjing, China) at 200× magnification.

### 2.5. Serum and Liver Biochemical Analysis

Serum and liver biochemical analyses were performed using the methods previously reported in Xu et al. [30] and Ren et al. [31] with some modifications. Total cholesterol (TC), triglyceride (TG), high-density lipoprotein (HDL-C), low-density lipoprotein (LDL-C), alanine aminotransferase (ALT), and aspartate aminotransferase (AST) were determined by Automatic biochemical analyzer (AU5800, Beckman, Brea, CA, USA). The commercial kits (Shanghai Enzyme-linked Biotechnology Co., Ltd., Shanghai, China) were used to measure the liver concentration of tumor necrosis factor-a (TNF-a), interleukin-6 (IL-6), monocyte chemoattractant protein-1 (MCP-1), and cyclooxygenase-2 (COX-2).

### 2.6. Gut Microbiota Analysis

The methods of DNA extraction and 16 s rRNA gene sequencing in rats feces were followed the method reported by Han et al. [32] with slight modifications. To accurately weigh 200-mg feces, QIAamp DNA Stool Mini Kit (Qiagen, Hilden, Germany) was used, and DNA of feces gene bacterial group was extracted according to the manufacturer’s instruction. V3 + V4 regions of bacterial 16 s rRNA (from 515 to 926,806) were amplified from the extracted DNA using bar-coded primers 515F (5′-GTGCCAGCMGCCGCGGTAA-3′) and 926R 806R (5′-GGACTACHVGGGTWTCTAAT CCGTCAATTCMTTTGAGTTT-3′). Sequencing was then performed on a MiSeq PE250 instrument (Illumina, San Diego, CA, USA). After the quality control of original data, sequences were assigned to operational taxonomic units (OTUs) by UPARES at 97% similarity. Next, alpha and beta diversity analysis were performed using QIIME. Significant differences in the relative abundance of microbial taxa between the HFC and HF groups were detected using linear discriminant analysis effect size (LEfSe).

### 2.7. Statistical Analysis

SPSS statistical software 24.0 was used to analyze all the data of this study. To compare the mean differences among the groups, one-way analysis of variance (ANOVA) was used. *p* < 0.05 was considered a significant difference. All values were expressed as the mean ± SD.

## 3. Results

### 3.1. The Bioactive Components of CPP

A total of 18 polyphenols was identified from CPP using a UPLC-MS/MS-based method. Narirutin was the most abundant phenolic component in CPP (Appendix A). In addition, CPP active ingredients contain polyphenols, essential oils, and pectin.

### 3.2. CPP Prevents Obesity Caused by High-Fat Diet

Weight gain is the most obvious characteristic of obesity. As shown in Figure 2A, there was no significant difference (*p* > 0.05) in the body weight of rats in the ND, HF, and HFC groups during the early part of the experiment, and the body weight of all rats increased over time. However, after the ninth week, rats in the HF group had a significant increase in body weight compared to the ND group (*p* < 0.05). Importantly, CPP addition significantly reduced weight gain in high-fat-fed rats (*p* < 0.05).

The precursor of obesity is the accumulation of fat in the liver. As shown in Figure 2B, at week 12, there was no significant difference in the increase in liver weight in the HF group compared to the ND group (*p* >0.05). However, CPP significantly reduced the increase in liver weight caused by HFD (*p* < 0.05). In addition, compared with ND group, the epididymis fat weight, perirenal fat weight, and subcutaneous fat weight of the HF increased, but these indicators in the HFC group were significantly lower than HF group (*p* < 0.05). These results suggested that CPP could prevent the increase of liver weight, epididymis fat weight, perirenal fat weight, and subcutaneous fat weight in HFD rats.

### 3.3. Histological Analysis

The protective effect of CPP on the liver was analyzed from a histological point of view. Figure 3 shows that the hepatocytes of HF group are scattered and perform balloon-like changes, the edges of the hepatic lobules are not clear, and the fatty deposits in the liver form more white fatty vacuoles. In ND group, the hepatocyte structure is complete and well-defined, with few vacuoles representing the location of oil. In addition, the characteristics of hepatocytes in the HFC and ND groups were similar, with no significant damage to hepatocytes in the HFC group and significantly fewer fat vacuoles than in the HF group. The results of histological observations showed that CPP was effective in reducing the accumulation of liver fat, which in turn had an anti-obesity effect.

The enlargement of fat cells leads to the growth of adipose tissue. In the HFD-rats, addition of CPP significantly reduced adipose tissues size and intracellular fat storage (Figure 3B).

### 3.4. The Anti-Inflammatory Effect of CPP

An increase in visceral fat leads to an increase in pro-inflammatory cytokines in the liver. Therefore, the concentrations of various well-defined pro-inflammatory cytokines in the liver were further determined to assess the effect of CPP on the inflammatory response in HFD rats. As shown in Figure 3C, the concentration of the pro-inflammatory cytokines in liver, such as IL-6, MCP-1, and COX-2, were significantly higher in the HF group compared to the ND group (*p* < 0.05). Obviously, elevated pro-inflammatory cytokines can be significantly reversed after supplementing CPP, including IL-6, MCP-1, and COX-2 (*p* < 0.05). The concentration of pro-inflammatory cytokines (TNF-α) in the HF group were not significantly different from those in the ND group (*p* > 0.05). However, after adding CPP to the HF group, the TNF-α concentration in the liver of the HFC group was significantly lower than those of the ND group (*p* < 0.05). These results suggest that CPP supplementation can reduce the inflammatory response induced by a HFD.

### 3.5. Effect of CPP on Serum Biochemical Parameters

We measured serum metabolic-related parameters that reflect liver function. The results for serum biochemical parameters, including total cholesterol (TC), triglycerides (TG), glucose (GLU), high-density lipoprotein cholesterol (HDL-C), low-density lipoprotein cholesterol (LDL-C), alanine aminotransferase (ALT), and aspartate aminotransferase (AST) levels, are shown in Figure 4. The levels of TC, TG, HDL-C, LDL-C, and ALT were significantly increased in the HF group compared to the ND group (*p* < 0.05), whereas HFD feeding had no significant effect on GLU and AST levels (*p* > 0.05). Supplementation with CPP significantly reduced serum levels of TC, TG, LDL-C, and ALT. However, supplementation with CPP had no significant effect on GLU and AST levels (*p* > 0.05). Furthermore, CPP supplementation resulted in a significant increase in HDL-C levels compared with the HF group (*p* < 0.05). In summary, CPP supplementation can improve abnormal lipid metabolism and protect the liver from damage caused by a high-fat diet.

### 3.6. CPP Modulated Gut Microbiota Diversity Induced by NAFLD

To evaluate the effect of CPP on the gut microbiota of HFD-fed rats, the bacterial abundance at the phylum and genus levels was detected by taxonomic analysis. As shown in Figure 5A, the 10 most abundant species at the phylum level in the three groups of samples. The top three phyla were *Firmicutes*, *Bacteroidota*, and *Campylobacterota*. Compared with the ND group, HFD treatment reduced the level of *Firmicutes* and *Bacteroidota* while increasing the level of *Campylobacterota*. However, the addition of CPP increased the level of *Firmicutes* in HFD-fed mice while reducing the level of *Campylobacterota*. The effects of the CPP on the gut microbiota structure of HFD-fed mice were further analyzed at the genus level, and Figure 5B shows the 30 most abundant species at the genus level in all samples. HFD-fed rats affected the abundance of *Faecalibaculum*, *Lactobacillus*, *Helicobacter*, *Blautia*, and *Bacteroides*. CPP supplementation remarkably increased the abundance of *Faecalibaculum* and *Lactobacillus* and reduced the abundance of *Helicobacter*, Blautia, and *Bacteroides*. Interestingly, *Dubosiella* were found in the HFC group but not in the ND and HF group.

PCoA analysis shows the sample distances between three groups (Figure 6). PCoA clearly distinguished the ND group from the HF and HFC group, and the HFC group was also clearly segregated from the HF group.

To identify the specific bacterial taxa after CPP supplementation, the LEfSe analysis with 3.0 as the threshold on the logarithmic LDA score for discriminative features was performed. Among the three groups, 63 phylotypes from phylum to species were discovered as high-dimensional biomarkers, and we found that the bacterial species in HF were abundant (Figure 7). These microbes mainly belonged to four different phyla, i.e., *Bacteroidota*, *Actinobacteriota*, *Desulfobacterota*, and *unidentified_Bacteria*. The genera *Faecalibaculum*, *Bifidobacterium*, and *Parabacteroides* were biomarkers in the ND group. The 10 genera (*Blautia*, *Lactococcus*, *Erysipelatoclostridium*, *A2*, *Romboutsia*, *Roseburia*, *Lachnopiraceae_UCG_004*, *Vagococcus*, *Tuzzerella*, and *Desulfofrigus*) were predominant in the HF group. Remarkably, *Dubosiella*, *Lachnospiraceae_NK4A136_group*, *Desulfovibrio*, and *Colidextribacter* were biomarkers in the HFC group. In conclusion, our results showed that CPP could modulate gut microbiota composition in HFD-fed rats rather dramatically.

## 4. Discussion

The present study demonstrated the preventive effect of CPP on NAFLD of mice. CPP reduces HFD-induced dyslipidemia and hepatic triglyceride accumulation and has a positive effect on liver inflammation. Meanwhile, CPP improves the intestinal flora disorder caused by HFD, which exhibits the potential to be an important functional active food or prebiotic.

CPP containing a variety of bioactive compounds (Appendix A), which may have an important role in the health of the organism. Citrus fruit peel is a valuable source of essential oils, pectin, and polyphenol [9]. Limonene, the main component of citrus essential oil, has been widely reported to have good antibacterial and antioxidant activity [33,34]. Pectin in citrus fruits has been documented to be effective in lowering blood cholesterol levels and also in improving intestinal inflammation [9,35]. Jiang et al. [36] showed that anti-obesity and reduction of blood lipid levels are representative activities of citrus products for the treatment of fat metabolism disorders and that polyphenolic compounds in citrus peels contribute to the beneficial activity of citrus products. We identified a total of 18 phenolic compounds at CPP (Appendix A) and identified narirutin as the major compounds. Narirutin is an influential bioactive compound in citrus fruits. Park et al. [37] showed that the narirutin fraction extracted from *Citrus unshiu* peel was effective in preventing lipid formation and inhibiting the production of pro-inflammatory cytokines, thereby reducing alcohol-induced liver injury. As expected, from the ninth week to the twelfth week, CPP supplementation significantly reduced the body and organ weight gain of HFD rats. Our findings are similar to some previous reports that pomelo peel and sweet orange peel and their polyphenolic extracts are effective in reducing body weight in obese mice [38,39]. To better understand the effect of the CPP on anti-NAFLD, further evaluation of the bioactive components in CPP is needed.

There is growing evidence that HFD-induced lipid accumulation leads to metabolic disturbances, a key factor in the progression of obesity and hyperlipidemia in NAFLD [40,41,42]. Samsudin et al. [38] showed that sweet orange peel powder significantly reduced TC, TG, and LDL-C levels but did not increase HDL-C levels, which is consistent with our results. In addition, CPP supplementation did not affect serum GLU concentrations. Therefore, CPP may inhibit adverse lipid metabolism by lowering serum levels of TC, TG, and LDL-C. To determine whether the reduction in fat mass was related to adipocyte morphology, adipose tissue was stained using the H&E method to visualize the cell size. Consistent with the expected results, the size of adipocytes in HFD-fed rats were significantly larger than that of normal low-fat diet rats. Notably, the adipocytes of CPP-treated rats were significantly smaller than those from the HFD-fed rats. The above results suggest that CPP supplementation may attenuate HFD-induced fat mass gain by reducing fat accumulation and adipocyte size.

Following the formation of excess lipid accumulation in the liver, lipid peroxidation is induced, and the oxidative stress process begins. Then, the production and release of pro-inflammatory cytokines, such as TNF-α and IL-6, further induce liver inflammation and exacerbate the progression of NAFLD to NASH [43]. We observed that CPP supplementation significantly reduced the increase in pro-inflammatory cytokines, such as TNF-α, IL-6, and MCP-1, caused by HFD. Furthermore, Liu et al. [9] suggested that citrus peels could exhibit potent anti-inflammatory activity by inhibiting inflammation-related enzymes (COX-2), which is consistent with our results. Our results showed that the COX-2 expression was significantly elevated in HFD-fed mice compared to normal diet mice, while CPP supplementation significantly reduced COX-2 expression and restored it to normal levels. It is well known that ALT and AST in serum are important indicators of liver function and liver injury [44]. In the present study, the inhibition of HFD-induced elevation of serum ALT activity was strong when CPP was supplemented in HFD-fed rats. However, CPP had no significant inhibitory effect on the HFD-induced increase in serum AST activity. In summary, CPP supplementation reduces pro-inflammatory cytokines, decreases the expression of inflammation-associated enzymes, as well as inhibits serum ALT activity, thereby reducing liver injury induced by HFD.

Gut microbiota is considered a major risk factor for the development of NAFLD [45]. However, the mechanism by which citrus peel alters intestinal flora and thus prevents and treats NAFLD has rarely been reported. Therefore, this research investigates for the first time whether the prevention and treatment of NAFLD by CPP is associated with changes in the intestinal microbiota. Xie and Halegoua-DeMarzio [46] showed that disruption of intestinal integrity and the inflammatory cascade response through cytokine release promote liver inflammation as a possible mechanism by which dysbiosis of the intestinal flora leads to the development of NAFLD. In addition, studies have reported the involvement of *Firmicutes* in maintaining the integrity of the intestinal barrier [47]. As shown in Figure 5 and Figure 7, CPP treatment increased the relative abundance of fecal *Firmicutes* at the phylum level. Thus, our results suggest that CPP may protect the integrity of the intestinal barrier by increasing the relative abundance of *Firmicutes*, which in turn reduces HFD-induced liver inflammation in rats.

Evidences that the reduction of the genus *Dubosiella* inverts the dysbiosis of the intestinal microbiota [48]. However, our data suggest that *Dubosiella* did not appear in the feces of normal diet rats and HFD-fed rats, while CPP supplementation significantly increased the relative abundance of *Dubosiella*. Similarly, the results of Guo, Cao, Fang, Guo, and Li [23] showed that intake of Ougan juice and lactic acid bacteria fermented Ougan juice had no inhibitory effect on *Dubosiella* and concluded that *Dubosiella* had an inhibitory effect on obesity. The genus *Lachnospiraceae_NK4A136_group* can prevent and treat acute colitis [49]. Some researchers have found that *Lachnospiraceae_NK4A136_group* is butyrate-producing bacteria, and butyrate protects the intestinal mucosa and reduces host inflammation [50,51]. In the present study, we observed a remarkable increase in the abundance of *Lachnospiraceae_NK4A136_group* in the HFC group, suggesting that CPP supplementation protects the intestinal mucosa and reduces the occurrence of inflammation. Li et al. suggested that *Desulfovibrio* can promote lipopolysaccharide (LPS) production and thus induce intestinal inflammation [52]. Chen et al. concluded that *Desulfovibrio* was not always associated with adverse health effects and that *Desulfovibrio* was negatively associated with BMI (body mass index), UA (uric acid), TG, and waist measurements, which are indicators of obesity or metabolic disorders [53]. Interestingly, CPP supplementation significantly increased the abundance of *Desulfovibrio* in the feces of HFD-fed rats (Figure 5), which is consistent with Chen’s research. Thus, CPP may reduce the body weight of rats by causing an increase in *Desulfovibrio* abundance. *Colidextribacter*, as one of the biomarkers in CPP-treated HFD-fed rats (Figure 7), could be an indicator of improved HFD-induced NAFLD in the intestine. However, limited information is available on *Colidextribacter*. Therefore, further studies are needed to fully assess the behavioral patterns of *Colidextribacter* in the intestine.

Intestinal flora and inflammation are the main features of NAFLD [54]. In this study, we demonstrated that the addition of CPP reduced HFD-induced obesity, regulated blood lipids, decreased liver injury and liver inflammation, and altered the intestinal flora composition of HFD mice. These results suggest that the gut microbiota may be involved in the effects of the CPP on lipid metabolism and NAFLD.

## 5. Conclusions

Overall, our study showed that dietary CPP intake has significant preventive and therapeutic effects on NAFLD in HFD-fed mice by attenuating liver and fat metabolism disorders, reducing liver injury, and inhibiting the release of hepatic inflammatory cytokines. Importantly, this study provides insight into changes in the composition and abundance of the gut microbiota and finds that CPP supplementation can reshape the composition of the disordered microbiota. It is suggested that dietary supplementation with CPP may be a new strategy for the prevention and treatment of NAFLD.

## Figures and Tables

**Figure 1 foods-10-03022-f001:**
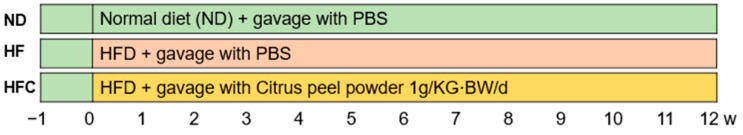
Experimental period diagram.

**Figure 2 foods-10-03022-f002:**
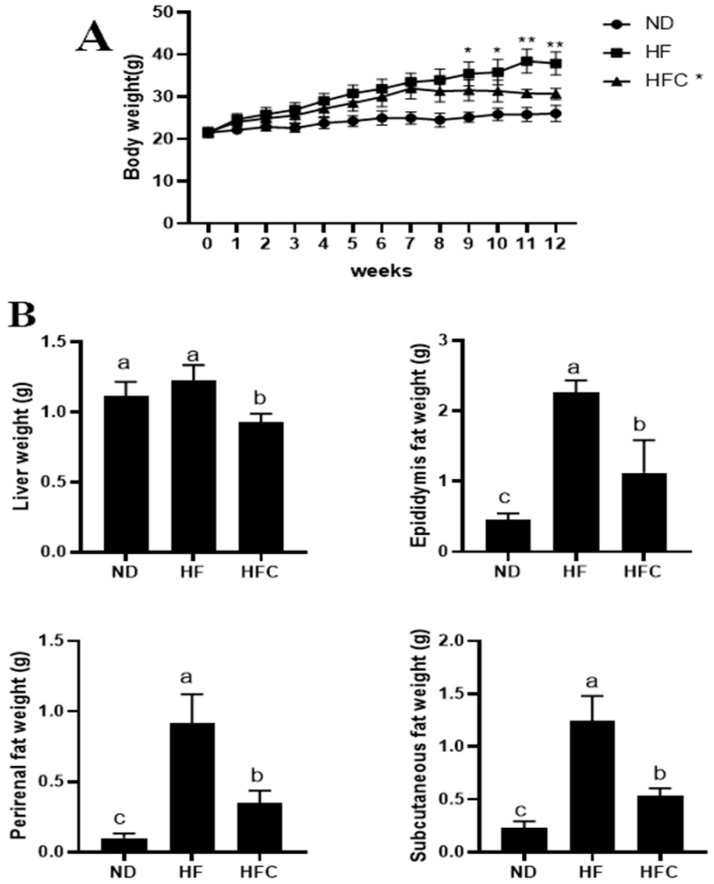
The effect of CPP administration on the physiological indexes and fat accumulation of HFD rats. Body weight growth of rats (**A**); * *p* < 0.05 and ** *p* < 0.01 compared with the HFC group. Liver weight, epididymal fat weight, perirenal weight, and subcutaneous fat weight of rats (**B**); means with different letters are significantly different (*p* < 0.05).

**Figure 3 foods-10-03022-f003:**
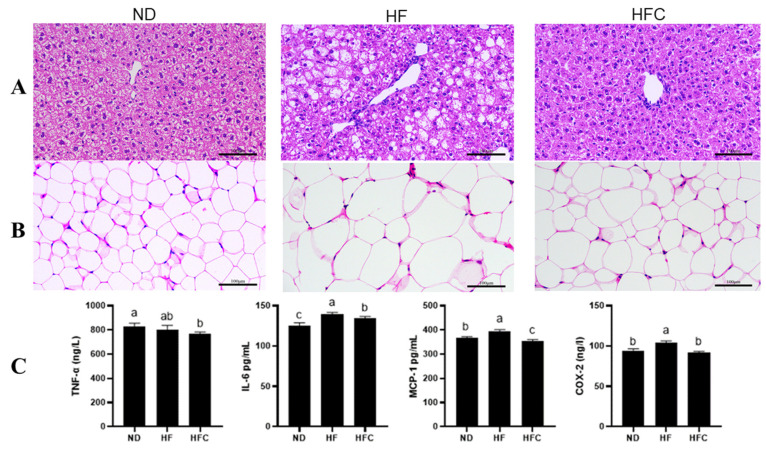
Effects of CPP administration on the liver and adipose histopathology and inflammation in rats fed a high-fat diet for 12 consecutive weeks. (**A**,**B**) Liver and eWAT sections stained with hematoxylin and eosin and representative images captured in 200×; (**C**) Hepatic inflammatory cytokines, including TNF-α, IL-6, MCP-1, and COX-2. Means with different letters are significantly different (*p* < 0.05).

**Figure 4 foods-10-03022-f004:**
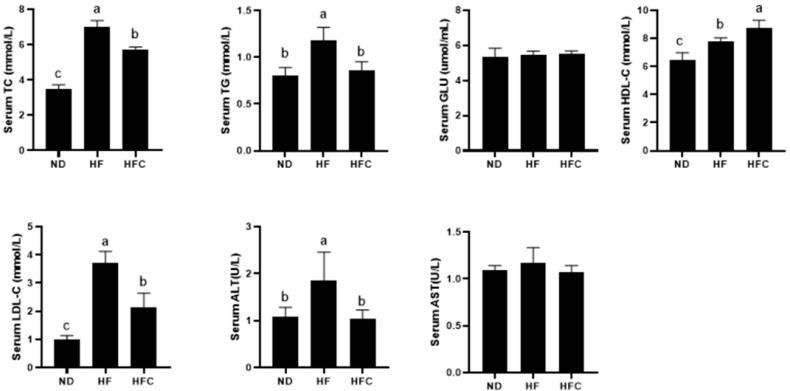
Effects of CPP supplementation on serum biochemical indexes (TC, TG, GLU, HDL-C, LDL-C, ALT, AST) in HFD rats for 12 consecutive weeks. Means with different letters are significantly different (*p* < 0.05).

**Figure 5 foods-10-03022-f005:**
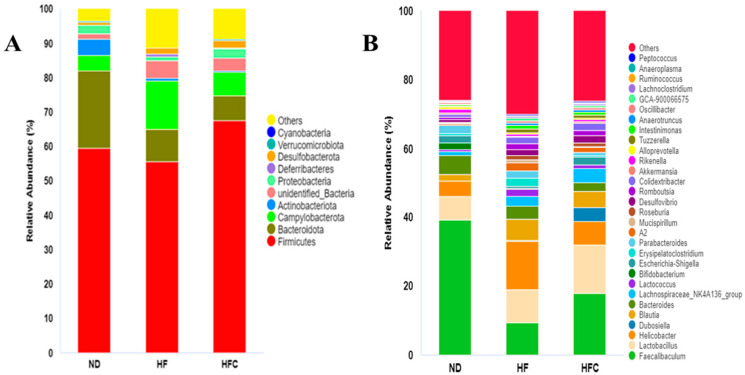
Composition of fecal microbiota at the phylum and genus levels. (**A**) Phyla (Top 10); (**B**) genera (Top 30).

**Figure 6 foods-10-03022-f006:**
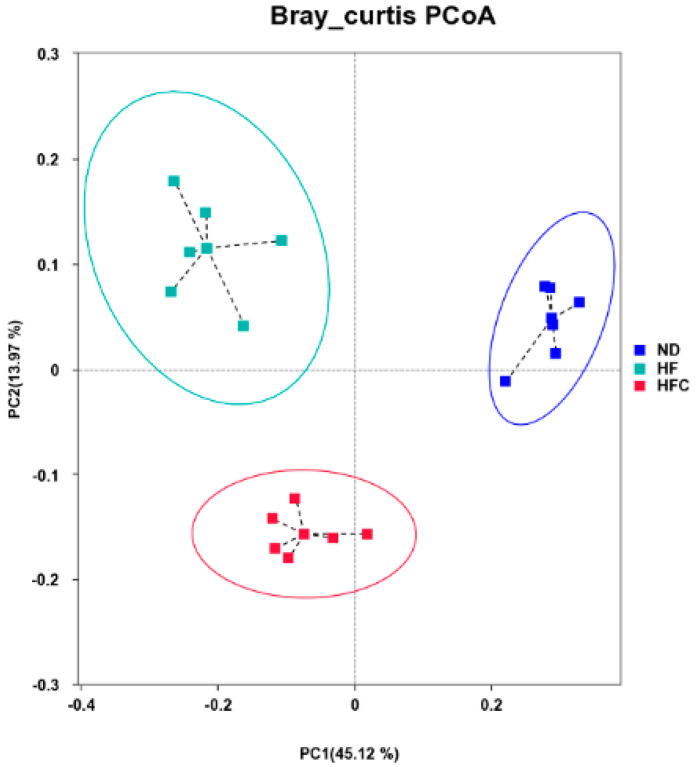
PCoA score plot of rat intestinal flora after addition of CPP to high-fat diets. Each point represents only one biological sample.

**Figure 7 foods-10-03022-f007:**
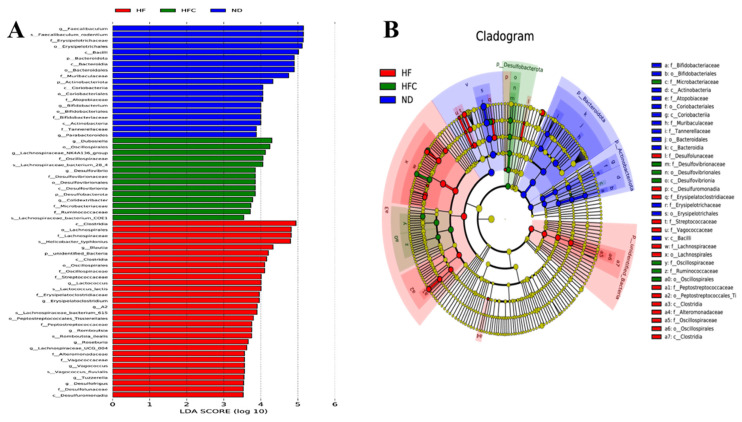
LEfSe analysis of intestinal flora. (**A**) LDA scores calculated for characteristics at the OTU level; (**B**) relative abundance of OTUs. Different colors indicate different groups. The colors indicate the major microbial biomarkers in the group, and the biomarker names are listed in the upper right corner.

## Data Availability

All data presented in this research are available through the corresponding author.

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
