# Peer review of "The Regulatory Effects of Citrus Peel Powder on Liver Metabolites and Gut Flora in Mice with Non-Alcoholic Fatty Liver Disease (NAFLD)"

_foods, 2021, doi:10.3390/foods10123022_

Round 1
Reviewer 1 Report
In the current study, authors aimed to explore new strategies and mechanisms associated with the consumption of CPP to alleviate nonalcoholic fatty liver injury, lipid metabolism disorders and gut microbiota dysbiosis in obese mice induced by high-fat diet (HFD). They came to the conclusion that CPP has a preventive effect on NAFLD, which can be linked to the regulation of the intestinal microbiota. This study is interesting and well designed. There are only minor points:
1- There are some grammatic mistakes. Please, revise the English language.
2- Animal design needs references for the protocol used in the study.
3- Figure 1 should be mentioned in its place in the text.
4- Authors should illustrate in methods how did they determined different weight types mentioned in figure 2.
5- What are symbols a,b, c on figures? It should be mentioned in figure legends.
6- In line 215, authors wrote "the TNF-α levels in the liver of the HFC group was significantly higher than those of the ND group". Please, revise this section because the figure 3 indicates a decrease in TNF in HFC group compared with ND group.
7- Line 228, they wrote "low- density lipoprotein cholesterol (HDL-C)". This should be corrected to "LDL-C"
Reviewer 2 Report
Meiyi Hu et al. reported that citrus peel powder improves the NAFLD in high-fat-fed obese mice by modulating the intestinal microflora.
- Citrus peel is known to be effective in obesity and NAFLD. Why was CPP, a complex of pomelo peel powder and orange peel powder, used to examine the effect on nonalcoholic fatty liver disease? Did CPP administration show a better effect on NAFLD than the pomelo peel powder and orange peel powder alone administration?
-
Line 117~124: Did you corn oil as a vehicle for the ND and HF groups? In Figure 1, it is presented that PBS was used as the vehicle. It needs to check this. If you used corn oil as a vehicle, what was the reason?
-
Line 170-173: CPP contains a large amount of narirutin and vanillic acid. Are these ingredients a bioactive compound?
-
Line 175~180: The decrease in weight gain appears after 9 weeks of CPP administration compared to the HF. It needs to explain whether it was not increased the gain weight from the 9th week after CPP administration or it was reduced the body weight at 12 week compared with at 9 week. It is necessary to discuss this phenomenon.
-
Line 206 and Fig 3C: Proinflammatory cytokine levels in the liver requires conversion to liver protein concentration. It needs to check this.
-
Line332-339: Although an anti-inflammatory effect was observed in the liver, it is necessary to determine the proinflammatory cytokine in the intestine and serum to explain the anti-inflammatory effect caused by the increase in intestinal microbiota Firmicutes. In Line311-313, the anti-inflammatory effect in liver tissue is explained by lipid peroxidation, oxidative stress, and proinflammatory cytokine. It is necessary to explain how changes in the gut microbiota are involved in inflammation in the liver.
-
The authors described that the intestinal microflora regulates the inflammatory response in the gut. A detailed explanation is needed on the relationship between gut microbiome, inflammation, and NAFLD. If the authors suggest that changes in the gut microbiota caused by CPP reduce the intestinal inflammatory response, thereby suppressing the onset of obesity and NAFLD, further discussion is needed in this.
-
It is reasonable to observe the improvement of gut microbiota and NAFLD by the administration of CPP. Do you think the decrease in NAFLD by CPP result from the changes in the intestinal microbial community?
Author Response
Dear Reviewer 2:
Thank you very much for sending our work (foods-1428786) for peer review and inviting us to revise our work. We appreciate the positive and encouraging comments from the reviewers. Those comments are all valuable and very helpful for revising and improving our paper, as well as the important guiding significance to our researches. We carefully studied the comments and revised the manuscript based on the reviewers' comments. The areas marked with highlights are modified. Our point-by-point responses are presented below.
- Thank you for your careful review of our manuscript. First of all, citrus fruits produce a large amount of byproducts every year causing waste and the risk of polluting the environment. The phenolic compounds in the peel of pomelo and sweet orange, the main citrus fruits, have been widely reported for their excellent health benefits. Secondly, we think that the fusion of two materials rich in phenolic sources may help to increase the variety and content of phenolic compounds, which in turn may lead to better health effects through synergistic effects. There has been much literature showing that phenolic compounds, mainly phenolic acids and flavonoids, in pomelo peel and sweet orange peel have good preventive and therapeutic effects on obesity. Importantly, obesity serves as an important factor in targeting NAFLD. Therefore, we believe that CPP also has the potential to be effective in the prevention and treatment of NAFLD.
- We thank the reviewer for this helpful suggestion. We have corrected this.
- Vanillic acid and narirutin are both bioactive compound. We have added explanations about these two components to the discussion (line 290-300).
- We agree with the reviewers’ suggestion. We have added an explanation of this phenomenon to the discussion (line 300-305).
- Per suggestion, we have corrected this issue in the text.
- Thank you for your suggestions. We have appropriately removed a portion of the reference citation due to its inappropriateness. Line 340-348: we describe in detail the relationship between CPP-promoted increases in the relative abundance of Firmicutes and liver inflammation.
- Per suggestion, we deleted the relevant content about CPP regulating the intestinal flora to reduce intestinal inflammation. In discussion, we focus on the relationship between gut microbiota, liver inflammation and NAFLD (line 373-379).
- Gut microbiota is a novel biomarker of NAFLD development, which promotes NAFLD development by activating inflammatory responses and regulating metabolic mechanisms [Aron-Wisnewsky, J., Vigliotti, C., Witjes, J., Le, P., Holleboom, A. G., Verheij, J., Clement, K. (2020). Gut microbiota and human NAFLD: disentangling microbial signatures from metabolic disorders. Nature Reviews Gastroenterology & Hepatology, 17, 279–297]. Therefore, I think that altered intestinal flora is the result of CPP reducing NAFLD. In other words, a decrease in NAFLD leads to a change in the intestinal flora.
If you have any queries, please don’t hesitate to contact me by email.
Sincerely yours,
Zheng Ruan.
